The effect of the erythropoietin resistance index on brain hemorrhage and infarction risk in maintenance hemodialysis patients: a retrospective cohort study

Liu Mingyu 1 2
Ren Kaiming 1
Wang Qian 1
Zhao Chen 1
Ren Zhuo 1
Bai Jiuxu 107034054@qq.com 1
Cao Ning bzxyjhk@126.com 1
1 Department of Blood Purification, General Hospital of Northern Theatre Command , Shenyang , Liaoning , China
2 Postgraduate College, Dalian Medical University , Dalian , Liaoning , China
Anson Lesley
Electronic publication date: 2025 Nov 12
Publication date: 2025
Volume: 13
Electronic Location ID: e20326
Received 2025 Mar 31; Accepted 2025 Oct 13
Copyright: ©2025 Liu et al.
Copyright year: 2025
Copyright holder: Liu et al.
License: This is an open access article distributed under the terms of the Creative Commons Attribution License, which permits unrestricted use, distribution, reproduction and adaptation in any medium and for any purpose provided that it is properly attributed. For attribution, the original author(s), title, publication source (PeerJ) and either DOI or URL of the article must be cited.
License URL: https://creativecommons.org/licenses/by/4.0/

Keywords: Maintenance hemodialysis, Erythropoietin, Brain hemorrhage and infarction

Funding: The Liaoning Provincial Science and Technology Plan Joint Fund 2023-MSLH-350 The 2024 Health and Family Planning Special Scientific Research Project Fund No. 24BJZ46 This research was funded by the Liaoning Provincial Science and Technology Plan Joint Fund (2023-MSLH-350), and the 2024 Health and Family Planning Special Scientific Research Project Fund (No. 24BJZ46). The funders had no role in study design, data collection and analysis, decision to publish, or preparation of the manuscript.

==============================
Background

Maintenance hemodialysis (MHD) is the main renal replacement therapy for patients with end-stage renal disease, and erythropoietin (EPO) is the main therapy for renal anemia in patients receiving maintenance hemodialysis. However, the risks of brain hemorrhage and infarction in patients on hemodialysis with erythropoietin therapy remain unclear.

Methods

Patients who underwent regular hemodialysis at the Blood Purification Center of the Northern Theater General Hospital from January 1, 2018, to January 1, 2022, were retrospectively selected. A total of 588 MHD patients were enrolled on the basis of the inclusion and exclusion criteria. The primary endpoint of follow-up was brain hemorrhage and infarction, or January 1, 2024. The secondary endpoint was all-cause death. The erythropoietin resistance index (ERI) was calculated as follows: erythropoiesis-stimulating agent (ESA) (IU/week)/body weight (kg)/hemoglobin level (g/dL). Patients were divided into four groups according to ERI quartile, and a Cox proportional risk model was applied to observe the associations between the ERI and the risks of brain hemorrhage and infarction and all-cause mortality.

Results

During a median follow-up of 6 years, brain hemorrhage events occurred in 41 (6.9%) MHD patients, and brain infarction events occurred in 61 (10.3%) MHD patients. According to the Kaplan–Meier curve, the incidence of brain hemorrhage and infarction was greater in the high-ERI subgroup than in the low-ERI subgroup (p < 0.001). Multivariate Cox regression analysis revealed that a high ERI was a significant predictor of brain hemorrhage (HR: 3.85, 95% CI [1.34–11.07], p = 0.012) and brain infarction (HR = 2.657, 95% CI [1.17–6.02], p = 0.020). A higher ERI was associated with an increased risk of all-cause death in MHD patients (HR: 1.72, 95% CI [1.05–2.82], p = 0.033).

Conclusions

Higher ERI scores were associated with brain hemorrhage and infarction in MHD patients.

Introduction

The number of hemodialysis patients is increasing annually. In the Chinese Dialysis Outcomes and Practice Patterns Study, Phase 5 (DOPPS5) study, the mean annual mortality rate for maintenance hemodialysis (MHD) patients was 8.8%, and cerebrovascular events were among the most common single causes of death (Xinju, 2022). Renal anemia is one of the most prevalent complications in MHD patients; it can reduce quality of daily life and may lead to an increased incidence of cardiovascular events and mortality (Abramson et al., 2003). Erythropoietin (EPO) is currently the primary treatment for renal anemia in patients on MHD, with 78.2% of MHD patients receiving EPO as of 2020 (GBD, 2020). The use of EPO has many benefits for patients with MHD, including improved quality of life and a reduced need for blood transfusion, but current studies have shown that the use of high doses of EPO may lead to an increased risk of stroke and all-cause mortality in MHD patients (Zhao et al., 2024; Suttorp et al., 2015).

Renal anemia cannot be corrected or improved with adequate or even excessive doses in some patients, and these patients may have EPO resistance or low EPO responsiveness (NKF, 2007). In recent years, the erythropoietin resistance index (ERI) has been proposed as an important evaluation index for evaluating the EPO response, and a high ERI indicates a poor EPO response (Okazaki et al., 2014). Using a high dose of EPO to reach the target hemoglobin level may increase the risk of stroke (Marto et al., 2021). Recently, a retrospective study revealed that a lower baseline response to erythropoietin can predict death, nonfatal myocardial infarction and nonfatal stroke (Mc Causland et al., 2025). However,the relationships between ERI and the risks of brain hemorrhage and infarction in MHD patients are still unclear. Our study was performed with the aim of exploring the associations between the ERI and brain hemorrhage and infarction and all-cause mortality in MHD patients.

Materials and Methods

Research design

In this retrospective study, we included 734 patients on MHD who underwent regular hemodialysis treatment at the Blood Purification Center of the General Hospital of Northern Theater Command from January 1, 2018, to January 1, 2022. The inclusion criteria were as follows: regular hemodialysis for more than 6 months involving dialysis 3 times/week for 4 hours/time; age >18 years; regular use of EPO during dialysis; other drugs, such as HIF inhibitors, which are not used to treat anemia; and complete dialysis data. The exclusion criteria were as follows: malignant tumors, cirrhosis, unexplained thrombocytopenia, blood disorders, and autoimmune diseases; surgery, active bleeding or blood transfusion, severe infection, chemotherapy, or immunosuppressant use within 3 months before the start of the study; and transfer to another dialysis center, change to peritoneal dialysis, kidney transplantation, etc. Patients whose data were incomplete were excluded. A total of 588 MHD patients were included according to the inclusion and exclusion criteria. All dialysis procedures were performed using identical hemodialysis equipment (F80, Fresenius, Bad Homburg vor der Höhe, Germany), with the ultrafiltration parameters and blood flow rates being determined by the treating physician according to individual patient characteristics including dry weight and clinical status.The primary endpoint of follow-up was brain hemorrhage and infarction, or January 1, 2024. The secondary endpoint was all-cause death. Brain hemorrhage and infarction were defined as acute injury of the central nervous system diagnosed on the basis of imaging, pathology, or other objective clinical evidence associated with clinical symptoms (Sacco et al., 2013).

This study was conducted in accordance with the principles set forth in the Declaration of Helsinki and was approved by the Clinical Research Ethics Committee of the Northern Theater General Hospital. Informed consent was exempt from the study, with approval number Y2024–204.

Data collection

We collected demographic data such as age; sex; body mass index (BMI); dialysis duration; type of primary disease; presence or absence of coronary heart disease, hypertension, diabetes mellitus (type 2 diabetes) or atrial fibrillation; and previous stroke history. Blood samples were collected from each patient in the early morning after overnight fasting and before hemodialysis began, and blood tests, including those for hemoglobin, C-reactive protein, alkaline phosphatase, total cholesterol, creatinine, blood urea nitrogen, parathyroid hormone, serum calcium, serum phosphorus, serum albumin (ALB), Kt/v, serum ferritin, transferrin saturation and β2 microglobulin, were performed. Treatment information, including iron supplementation, drug combinations and weekly mean values (IU/week) of the total ESA dose six months after the patient was enrolled, was collected. The ERI was calculated with the following formula: erythropoiesis-stimulating agent (ESA) (IU/week)/body weight (kg)/hemoglobin level (g/dL). The primary kidney disease types were classified as chronic glomerulonephritis, diabetic nephropathy, hypertensive nephropathy, or other causes. Patients were divided into four groups according to the ERI quartile.

Data and analysis

The measurement data are expressed as the mean ± standard deviation (x ± s), the measurement data that did not conform to a normal distribution are expressed as the median (25%, 75%), and the count data are expressed as the number of cases (n (%)). With respect to comparisons between groups, single-factor anaylsis of variance (ANOVA) was used to determine the differences between continuous variables, the Kruskal–Wallis H test was used for nonparametric data, and the chi-square test was used for categorical variables. The Kaplan–Meier method was used to compare the incidence of brain hemorrhage and infarction events and mortality among all groups, and the log-rank test was used to test the difference between survival curves. A multifactor Cox model was used to calculate the incidence of brain hemorrhage and infarction events and all-cause mortality in patients stratified by ERI quartile and in regular hemodialysis patients with end-stage renal disease.

By adding different covariates step by step, a multivariate Cox regression model stratified by ERI quartile was established. The models were specified as follows: (i) Model adjusted for demographics: age, sex, BMI, dialysis duration, and type of primary kidney disease; (ii) model adjusted for case mix: additional adjustment for comorbidities Iron supplementation and drug combinations; (iii) final model: additional adjustment for C-reactive protein, alkaline phosphatase, total cholesterol, creatinine, blood urea nitrogen, parathyroid hormone, serum calcium, serum phosphorus, and serum albumin, Kt/v; and serum ferritin, transferrin saturation and β2 microglobulin.

IBM SPSS 25.0 (IBM Corp., Armonk, NY, USA) was used for statistical analysis, and p < 0.05 was considered statistically significant.

Results

Baseline characteristics

Data were collected from 588 patients from the Blood Purification Center at the Northern Theater General Hospital. The mean age was 53.6 ± 12.3 years, and 59.9% were male. The primary diseases included hypertension (26.3%), diabetes (22.4%), chronic glomerulonephritis (17.8%) and others (33.3%). At the end of follow-up, 41 patients had experienced brain hemorrhage (6.9%), 61 patients had experienced brain infarction (10.3%), and 152 patients had died (25.8%). Patients were divided into four groups according to the ERI quartile: Q1: ERI ≤ 7.26, Q2: 7.26 < ERI ≤ 10.30, Q3: 10.30 < ERI ≤ 14.33 and Q4: ERI ≥ 14.33.

Demographic characteristics and laboratory data were compared between the groups of patients stratified by ERI quartile (Table 1).

Table 1 Demographic characteristics and laboratory measurements.

	ERI ≤ 7.26 (n = 146)	7.26 < ERI ≤ 10.30 (n = 151)	10.30 < ERI ≤ 14.33 (n = 145)	ERI ≥ 14.33 (n = 146)	p value					
Age, years	52.15 ± 12.74	53.55 ± 11.38	54.89 ± 13.11	53.97 ± 12.05	0.293					
Sex, male, n (%)	97 (66.4%)	91 (60.3%)	80 (55.2%)	84 (57.5%)	0.231					
Dialysis duration, years	4.19 (1.00, 8.11)	2.94 (0.50, 6.17)	3.00 (0.50, 6.30)	2.17 (0.50, 5.36)	0.002					
BMI, kg/m2	23.2 ± 3.37	23.23 ± 3.42	22.99 ± 3.89	22.95 ± 3.90	0.867					
Type of primary kidney disease, n (%)					0.492					
Hypertension	36 (24.7%)	35 (23.2%)	44 (30.3%)	40 (27.4%)						
Diabetes	31 (21.2%)	31 (20.5%)	36 (24.8%)	34 (23.3%)						
Chronic glomerulonephritis	24 (16.4%)	32 (21.1%)	28 (19.3%)	21 (14.4%)						
Others/unknown	55 (37.7%)	53 (35.1%)	37 (25.5%)	51 (34.9%)						
Coronary heart disease, n (%)	44 (30.1%)	40 (26.5%)	46 (31.7%)	43 (29.5%)	0.794					
Hypertension, n (%)	94 (64.4%)	115 (76.2%)	111 (76.6%)	119 (81.5%)	0.007					
Diabetes mellitus, n (%)	35 (24.0%)	42 (27.8%)	41 (28.3%)	46 (31.5%)	0.557					
Atrial fibrillation, n (%)	18 (12.3%)	16 (10.6%)	19 (28.8%)	13 (19.7%)	0.673					
History of stroke, n (%)	17 (11.6%)	20 (13.2%)	16 (11.0%)	16 (11.0%)	0.922					
Aspirin, n (%)	21 (14.4%)	19 (12.6%)	30 (20.7%)	18 (12.3%)	0.155					
ACEI/ARB, n (%)	76 (52.1)	98 (64.9%)	97 (66.9%)	82 (56.2%)	0.028					
EPO/w,IU	3, 875 (3,062.00, 5,000.00)	6, 625 (5,875.00, 7,375.00)	8, 750 (7,854.17, 9,750.00)	10,458.33 (9,552.08, 12,645.83)	<0.001					
ERI	5.19 (4.13, 6.47)	8.83 (7.97, 9.65)	12.48 (11.27, 13.48)	17.80 (15.87, 21.93)	<0.001					
Hb, g/dL	11.83 ± 1.43	11.36 ± 1.35	10.77 ± 1.18	9.49 ± 1.66	<0.001					
Alb,g/L	39.00 (37.28, 41.53)	38.70 (36.60, 40.80)	38.00 (35.70, 41.05)	37.40 (34.78, 40.20)	0.003					
CRP, mg/L	2.50 (1.20, 10.30)	3.40 (1.30, 12.98)	2.30 (1.30, 10.31)	3.51 (1.30, 14.21)	0.440					
AKP,IU/L	67.55 (53.99, 87.78)	71.54 (52.57, 92.04)	67.98 (49.94, 94.87)	70.77 (50.39, 105.53)	0.694					
Tcho, mmol/L	4.30 (3.60, 4.99)	4.30 (3.67, 5.29)	4.10 (3.47, 4.86)	4.09 (3.50, 4.92)	0.146					
Cr,μmol/L	1,136.41 ± 28.68	1,050.07 ± 26.89	986.30 ± 25.95	977.01 ± 29.97	<0.001					
BUN, mmol/L	21.20 (21.96, 30.08)	24.89 (21.36.2976)	25.63 (21.43, 30.62)	25.88 (21.75, 31.46)	0.698					
PTH, ng/L	289.50 (133.00, 634.00)	232.00 (118.00, 507.00)	231.00 (113.00, 486.50)	322.50 (166.25, 618.75)	0.043					
Ca, mmol/L	2.23 ± 0.019	2.23 ± 0.018	2.20 ± 0.019	2.15 ± 0.020	0.009					
P, mmol/L	1.97 ± 0.046	1.94 ± 0.053	1.90 ± 0.047	1.92 ± 0.045	0.746					
Kt/v	1.27 (1.07, 1.45)	1.27 (1.12, 1.53)	1.33 (1.16, 1.50)	1.28 (1.15, 1.48)	0.304					
Ferritin, ng/mL	220.95 (73.54, 536.25)	230.90 (88.80, 648.50)	222.30 (85.91, 539.15)	303.20 (105.65, 798.03)	0.128					
TSAT (%)	32.66 ± 12.09	31.88 ± 11.64	31.78 ± 14.45	30.77 ± 12.88	0.657					
Iron supplementation, n (%)	9 (6.2%)	9 (6.0%)	15 (10.3%)	17 (11.6%)	0.190					
β2-MG, mg/L	27.17 ± 6.79	27.16 ± 7.64	27.07 ± 7.29	27.76 ± 7.03	0.990					
Notes.

BMI body mass index

ESRD end-stage renal disease

ACEI/ARB angiotensin-converting enzyme inhibitor/angiotensin II receptor blocker

WBC leukocyte

Alb albumin

EPO erythropoietin

ERI erythropoiesis resistance index

CRP C-reactive protein

AKP alkaline phosphatase

Tcho total cholesterol

Cr creatinine

BUN blood urea nitrogen

PTH parathyroid hormone

Ca serum calcium

P serum phosphorus

Kt/v dialysis effectiveness index

TSAT transferrin saturation

β2-MG β2-microglobulin

KM curve

The Kaplan−Meier survival curves for each group revealed that a higher ERI was significantly associated with a greater risk of brain hemorrhage (log-rank test: p = 0.002, Fig. 1) and brain infarction (log-rank test: p = 0.004, Fig. 2) and worse survival (log-rank test: p = 0.021, Fig. 3).

Figure 1 KM curve of ERI quartile-grouped brain hemorrhage.

The 588 patients were divided into four groups according to the quartile of the baseline ERI. KM survival analysis of brain hemorrhage in the four groups revealed that patients in the higher ERI groups had a higher rate of brain hemorrhage than those in the lower ERI groups did (log-rank test, p = 0.002).

Figure 2 KM curve of ERI quartile-grouped brain infarction.

The 588 patients were divided into four groups according to the quartile of the baseline ERI. KM survival analysis of brain infarction in the four groups revealed that patients in the higher ERI group had a higher rate of brain infarction than those in the lower ERI groups did (log-rank test, p = 0.004).

Figure 3 KM curve of ERI quartile-grouped all-cause mortality.

The 588 patients were divided into four groups according to the quartile of the baseline ERI. KM survival analysis of all-cause mortality in the four groups revealed that patients in the higher ERI groups had a higher death rate from all-cause mortality than those in the lower ERI groups did (log-rank test, p = 0.021).

COX model

A multivariate Cox regression model analysis of the correlation between the ERI and brain hemorrhage incidence is shown in Table 2. The final model revealed that high ERIs were independent predictors of brain hemorrhage (HR: 3.85; 95% CI [1.34–11.07], p = 0.012).

A multivariate Cox regression model analysis of the correlation between the ERI and brain infarction incidence is shown in Table 3. The final model revealed that high ERIs were significantly associated with brain infarction (HR = 2.65, 95% CI [1.17–6.02], p = 0.020).

A multivariate Cox regression model analysis of the correlation between the ERI score and all-cause mortality is shown in Table 4. The final model revealed that high ERIs were independent predictors of all-cause mortality (HR: 1.72 95% CI [1.05–2.82], p = 0.033).

Discussion

In this study, we found that a higher ERI was associated with increased risks of brain hemorrhage and infarction, findings that remained relevant after adjusting for confounding factors such as comorbidities and drug use. Patients with higher ERIs may be more prone to brain hemorrhage and infarction events and at a greater risk of death. The ERI, an indicator used to evaluate EPO treatment responsiveness, can predict the occurrence of brain hemorrhage and infarction and all-cause death in MHD patients.

Patients on MHD have a 5–30 times greater risk of cardiovascular and cerebrovascular events than the general population does, and MHD patients may have a higher risk of death due to a stroke event than other patients do (Toyoda et al., 2005). A prospective multicenter study in Australia revealed that patients with end-stage renal disease had a higher mortality rate from stroke than did the general population (De La Mata et al., 2019). Yahalom et al. (2009) estimated the associations between the GFR and CKD and 1-year prognosis through a prospective study and concluded that the presence of a low eGFR combined with CKD is an independent predictor of mortality and poor prognosis in patients with acute stroke; this finding may be related to the increased risk of atherosclerosis in patients with end-stage renal disease. In addition, patients suffer from poor health due to common complications such as anemia and infection, thus increasing the mortality rate of stroke.

The 2012 edition of the KDIGO Guidelines states that caution should be taken when treating MHD patients with a history of stroke with EPO. In the placebo-controlled and double-blind trial of darbepoetin for type 2 diabetes and CKD (TREAT) conducted by Pfeffer et al. (2009) the stroke rate in the high-hemoglobin group was 5.0% in patients with CKD and 2.6% in the placebo group, indicating that the risks associated with treating patients with a history of stroke with EPO are considerable. A case-control study in the United States, after adjusting for multiple confounding variables, revealed that patients treated with EPO had a 30% greater risk of stroke than those not treated with EPO (Seliger et al., 2003). This may be due to the increased risk of cerebral hemorrhage in MHD patients with elevated blood pressure due to high doses of EPO, as well as the increased incidence of cerebral infarction due to the increased hemoglobin level and increased blood viscosity caused by EPO (Rodriguez et al., 2009). In recent years, a multicenter retrospective study using multivariate Cox regression analysis to adjust for confounders concluded that a higher ERI was associated with an increased incidence of cerebral hemorrhage in MHD patients; however, the association between the ERI and cerebral infarction incidence was not significant (Uchida et al., 2023). A retrospective study in Taiwan came to a different conclusion, noting that EPO treatment in hemodialysis patients was not associated with an increased risk of stroke or any of its subtypes. Although low doses of EPO are associated with an increased risk of ischemic stroke, the association is not dose dependent (Hung et al., 2021). Experimental studies have suggested that erythropoietin has a neuroprotective effect and can promote neurological recovery after stroke (Jerndal et al., 2010). In our study, a higher ERI was associated with the risk of both ischemic and hemorrhagic stroke in MHD patients after adjusting for multiple variables via a Cox model. High doses of EPO, which are needed to achieve higher target hemoglobin levels, may be associated with an increased risk of brain hemorrhage, infarction and other events in MHD patients without any associated significant improvement in quality of life (Singh et al., 2006).

Table 2 Hazard ratios for the incidence of brain hemorrhage by quartile of the ERI.

	Model 1		Model 2		Model 3		
	HR (95% CI)	p	HR (95% CI)	p	HR (95% CI)	p	
Q1:ERI ≤ 7.26	1.0 (reference)	0.014	1.0 (reference)	0.009	1.0 (reference)	0.015	
Q2:7.26 < ERI ≤ 10.30	0.91 (0.26, 3.15)	0.816	1.03 (0.29, 3.64)	0.958	0.97 (0.27, 3.45)	0.956	
Q3:10.30 < ERI ≤ 14.33	2.56 (0.90, 7.26)	0.078	2.60 (0.91, 7.39)	0.074	2.34 (0.81, 6.78)	0.119	
Q4:ERI ≥ 14.33	3.42 (1.26, 9.30)	0.016	3.94 (1.44, 10.81)	0.008	3.85 (1.34, 11.07)	0.012	
Notes.

Model 1 was adjusted for age, sex, BMI, dialysis duration, and type of primary kidney disease.

Model 2 was adjusted for age, sex, BMI, dialysis duration, type of primary kidney disease, comorbidities, iron supplementation and drug combinations.

Model 3 was adjusted for age, sex, BMI, dialysis duration, type of primary kidney disease, and comorbidities, Iron supplementation and drug combinations, C-reactive protein, Alkaline phosphatase, Total cholesterol, creatinine, Blood urea nitrogen, Parathyroid hormone, serum calcium, serum phosphorus, serum albumin, Kt/v; serum ferritin, transferrin saturation and β2Microglobulin.

Table 3 Hazard ratios for the incidence of brain infarction by quartile of the ERI.

	Model 1		Model 2		Model 3		
	HR (95% CI)	p	HR (95% CI)	p	HR (95% CI)	p	
Q1:ERI ≤ 7.26	1.0 (reference)	0.017	1.0 (reference)	0.003	1.0 (reference)	0.017	
Q2:7.26 < ERI ≤ 10.30	1.23 (0.53, 2.83)	0.627	1.14 (0.48, 2.70)	0.767	1.04 (0.43, 2.51)	0.927	
Q3:10.30 < ERI ≤ 14.33	1.37 (0.59, 3.15)	0.465	1.22 (0.52, 2.85)	0.643	1.05 (0.43, 2.58)	0.922	
Q4:ERI ≥ 14.33	2.74 (1.30, 5.80)	0.008	3.10 (1.44, 6.66)	0.004	2.65 (1.17, 6.02)	0.020	
Notes.

Model 1 was adjusted for age, sex, BMI, dialysis duration, and type of primary kidney disease.

Model 2 was adjusted for age, sex, BMI, dialysis duration, type of primary kidney disease, comorbidities, iron supplementation and drug combinations.

Model 3 was adjusted for age, sex, BMI, dialysis duration, type of primary kidney disease, and comorbidities, Iron supplementation and drug combinations, C-reactive protein, Alkaline phosphatase, Total cholesterol, creatinine, Blood urea nitrogen, Parathyroid hormone, serum calcium, serum phosphorus, serum albumin, Kt/v; serum ferritin, transferrin saturation and β2Microglobulin.

Table 4 Hazard ratios for the incidence of death by quartiles of the ERI.

	Model 1		Model 2		Model 3		
	HR (95% CI)	p	HR (95% CI)	p	HR (95% CI)	p	
Q1:ERI ≤ 7.26	1.0 (reference)	0.037	1.0 (reference)	0.015	1.0 (reference)	0.127	
Q2:7.26 < ERI ≤ 10.30	1.28 (0.78, 2.09)	0.333	1.26 (0.76, 2.08)	0.373	1.16 (0.70, 1.94)	0.571	
Q3:10.30 < ERI ≤ 14.33	1.37 (0.84, 2.24)	0.209	1.25 (0.76, 2.05)	0.383	1.15 (0.69, 1.90)	0.597	
Q4:ERI ≥ 14.33	1.93 (1.21, 3.08)	0.006	2.03 (1.27, 3.24)	0.003	1.72 (1.05, 2.82)	0.033	
Notes.

Model 1 was adjusted for age, sex, BMI, dialysis duration, and type of primary kidney disease.

Model 2 was adjusted for age, sex, BMI, dialysis duration, type of primary kidney disease, comorbidities, iron supplementation and drug combinations.

Model 3 was adjusted for age, sex, BMI, dialysis duration, type of primary kidney disease, and comorbidities, Iron supplementation and drug combinations, C-reactive protein, Alkaline phosphatase, Total cholesterol, creatinine, Blood urea nitrogen, Parathyroid hormone, serum calcium, serum phosphorus, serum albumin, Kt/v; serum ferritin, transferrin saturation and β2Microglobulin.

A previous study suggested a possible relationship between higher ESA doses and a greater risk of death (Streja et al., 2016), and a retrospective single-center study revealed that higher ESA doses were associated with an increased risk of all-cause death (Weinhandl, Gilbertson & Collins, 2011). However, a US study in which a marginal structure model was used to adjust for confounding factors revealed that the EPO dose was not associated with the risk of death in MHD patients (Ouhong et al., 2010). Bradbury et al. (2009) conducted a retrospective study of patients with persistently low hemoglobin levels below 11 g/dL and concluded that in patients with persistently low Hb levels, the risk of death was strongly associated with the patient’s hematopoietic ability but not the EPO dose. The ERI, an indicator of EPO responsiveness in MHD patients, has been considered a useful indicator for evaluating EPO resistance in recent years (Kaysen et al., 2006; Pan et al., 2022). Current studies have shown that a higher ERI is associated with increased all-cause mortality (Eriguchi et al., 2015; López-Gómez, Portolés & Aljama, 2008). After applying propensity score adjustment by Hayashi et al. (2019) multivariate Cox regression still revealed that a higher ERI was still associated with all-cause mortality. Fujikawa et al. (2012) evaluated 2,104 HD patients in the DOPPS study and suggested that persistently high ERI values were associated with elevated all-cause mortality in MHD patients. Lu et al. (2020) reported that serum ALB, high-sensitivity C-reactive protein, ferritin and creatinine before dialysis were independent factors related to EPO responsiveness in maintenance hemodialysis patients. Patients with higher ERIs had higher all-cause mortality and cardiovascular mortality. Fujii et al. (2023) conducted a 24-month prospective study on 4,034 patients and reported that the baseline ERI could predict all-cause mortality events but not cardiovascular events. However, the time-dependent ERI can predict the risk of all-cause mortality and cardiovascular events. A retrospective study in the United States that used a Cox proportional risk model to adjust for factors such as dialysis adequacy and serum ALB and ferritin concentrations demonstrated that a higher ERI and higher doses of EPO were associated with increased mortality (Nishio et al., 2013). However, Ogawa et al. (2014) through a 12-month cohort study, concluded that higher ESA doses and lower Hb levels were associated with an increased risk of all-cause mortality, whereas the ERI was not. At present, the relationship between EPO and the risk of all-cause death in MHD patients is still unclear. We cannot determine the ideal therapeutic dose of EPO in MHD patients, and the management of hemoglobin levels in MHD patients remains an important clinical issue that needs to be addressed. This is one of the few clinical studies in which brain hemorrhage and infarction were used as endpoints, with a focus on the relationships between ERIs and brain hemorrhage and infarction in MHD patients. After 6 years of follow-up in this study cohort, the ERI was associated with an increased risk of all-cause death in MHD patients.

Our research indicates that the erythropoietin resistance index (ERI) is not only a biomarker for the treatment response of anemia in maintenance hemodialysis (MHD) patients but also a predictor of stroke and all-cause mortality risk. In clinical practice, the ERI can serve as a risk stratification tool to help identify high-risk individuals who require enhanced monitoring. The incidence of stroke events was significantly greater in the high-ERI group than in the low-ERI group, which may be related to the procoagulant state induced by high-dose erythropoietin (EPO) and poor blood pressure control (Jean-Baptiste et al., 2022). On the basis of the conclusions of this study, a persistently elevated ERI > 14.3 IU/kg/g/dL can be used as an indicator to initiate intensified cerebrovascular protection measures, including blood pressure monitoring and treatment plan adjustments (Caprio & Sorond, 2019). Moreover, a high ERI indicates that a comprehensive assessment of anemia management in MHD patients is necessary, including correcting iron deficiency, controlling the microinflammatory state, ensuring adequate dialysis, treating secondary hyperparathyroidism, etc. (Santos et al., 2020). For patients with high ERIs, drugs other than EPO can be used to increase the hemoglobin level of MHD patients and prevent the occurrence of cardiovascular and cerebrovascular diseases caused by the side effects of high-dose EPO application. Proline hydroxylase (HIF-PHD) inhibitors can currently be used as a treatment for renal anemia in EPO-resistant patients, and their ability to increase hemoglobin levels is comparable to that of EPO-α (Fishbane et al., 2022). Studies have shown that sodium–glucose cotransporter 2 (SGLT2) inhibitors can increase the hemoglobin level of patients and reduce the risk of major cardiovascular events (Packer, 2024). Currently, the efficacy of SGLT2 for treating renal anemia in MHD patients is worthy of attention. The calculation method of the ERI integrates information on ESA dosage, body weight, and hemoglobin, providing multidimensional prognostic predictive value and offering an objective basis for the precise treatment of renal anemia, thereby improving the quality of life and long-term prognosis of MHD patients.

This study has certain limitations. First, this was a retrospective study, and the information was collected mainly from the medical records system of our hospital and via telephone follow-up; thus, there may be some errors. Second, the EPO dose recorded at the time was the average weekly EPO dose for six months after the patient was enrolled, but the EPO dose applied fluctuated with hemoglobin level. Finally, patients’ data may have changed during follow-up; however, we used baseline data to define the exposure categories, masking the effect of changes in the levels.

Conclusions

Our study revealed that a high ERI was associated with brain hemorrhage and infarction in MHD patients. Patients in the high-ERI group had a significantly lower survival rate than those in the low-ERI group. Multivariate Cox regression analysis revealed that a high ERI was a predictor of brain hemorrhage and infarction. Patients with higher ERI scores also had an increased risk of all-cause mortality.

Supplemental Information

Supplemental Information 1 Raw data

All the original data of this manuscript

Supplemental Information 2 Codebook

The meaning corresponding to 1 and 2 in the data

Supplemental Information 3 STROBE Checklist

Thank you to all the physicians and patients who provided data to support this study.

Additional Information and Declarations

Competing Interests

Author Contributions

Human Ethics

Data Availability

The authors declare there are no competing interests.

Mingyu Liu conceived and designed the experiments, performed the experiments, analyzed the data, prepared figures and/or tables, authored or reviewed drafts of the article, and approved the final draft.

Kaiming Ren performed the experiments, prepared figures and/or tables, and approved the final draft.

Qian Wang performed the experiments, prepared figures and/or tables, and approved the final draft.

Chen Zhao performed the experiments, authored or reviewed drafts of the article, and approved the final draft.

Zhuo Ren performed the experiments, authored or reviewed drafts of the article, and approved the final draft.

Jiuxu Bai conceived and designed the experiments, performed the experiments, authored or reviewed drafts of the article, and approved the final draft.

Ning Cao conceived and designed the experiments, performed the experiments, authored or reviewed drafts of the article, and approved the final draft.

The following information was supplied relating to ethical approvals (i.e., approving body and any reference numbers):

This study was conducted in accordance with the principles set forth in the Declaration of Helsinki and was approved by the Clinical Research Ethics Committee of the Northern Theater General Hospital, with approval number Y2024–204.

The following information was supplied regarding data availability:

The raw measurements are available in the Supplemental File.

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
