# Peer review of "The effect of the erythropoietin resistance index on brain hemorrhage and infarction risk in maintenance hemodialysis patients: a retrospective cohort study"

_PeerJ, doi:10.7717/peerj.20326_

## Round 0.1 · original submission · Major Revisions

· Academic Editor

Major Revisions

Reviewer 1 ·

Basic reporting

no comment

Experimental design

no comment

Validity of the findings

no comment

Additional comments

This study is a retrospective cohort study on the association between erythropoiesis resistance index (ERI) and brain hemorrhage, brain infarction, and all-cause mortality in hemodialysis patients and is considered to be highly interesting.

Below are several points that I would like to confirm:

<1> Major Points for Clarification

①Since the unit of hemoglobin (Hb) used for ERI is g/dL, the typical range is considered to be 8–12 g/dL. However, based on the raw data, the digit scale appears to be large, which may explain why the ERI values in this study appear smaller compared to other studies. Could you please confirm whether the number of digits for Hb is correct?

②It would be advisable to include the number of subjects and the Hb values for each group in Table 1.

③In Model 1 of the multivariable Cox regression analysis, comparisons are made among groups stratified by ERI; therefore, it is not necessary to include ERI as a covariate in the adjustment variables.

<2> Minor Points for Clarification

①Line 83: Should "dialysis enter" be corrected to "dialysis center"?

②Reference No. 21: The first author name appears to be missing—could you please confirm?

Reviewer 2 ·

Basic reporting

The manuscript is well written. However, several sentences contain grammatical or structural issues. For example,
-Line 45-50: These three sentences are repetitive in terms of phrasing and could be simplified.
-Line 237: "Patients in the high-ERI group had a significantly lower survival rate than those in the low-ERI group did." The phrasing is awkward; "did" is unnecessary here.
-Figure legends: These are minimal and lack sufficient context or clarity.

The literature referencing is largely sufficient but could be improved. For example, in the Introduction (lines 54-71) the authors explain the burden of cerebrovascular events in MHD and the role of EPO. However, they do not cite any literature after 2021.

The article structure follows the conventional format and presents the aims, methods and outcomes clearly. The tables are well formatted and include appropriate clinical/lab parameters by ERI quartile. However, the figure legends are minimal with vague titles (e.g., Figure 1. "KM survival analysis of brain hemorrhage in the four groups"). The legends should state the sample sizes and ERI group definitions (quartiles). While the manuscript states (line 264) that "datasets are available upon reasonable request," PeerJ typically expects open sharing.

The study aim is clearly stated in lines 70–71: to explore the association between ERI and brain hemorrhage/infarction. The methods in lines 72–131 align with the aim, and ERI is calculated and stratified appropriately (line 106). The outcomes (hemorrhage, infarction, mortality) are directly related to the aims. Multivariate Cox models and survival curves are used effectively to test associations. The study limitations (lines 227–233) are acknowledged. However, linking the results back to clinical implications (e.g., threshold-based interventions) would improve the final interpretation.

Experimental design

This is an original retrospective cohort study evaluating the association between ERI and cerebrovascular outcomes in MHD patients. It aligns well with PeerJ’s scope in clinical nephrology and contributes novel findings based on a sizable cohort (n = 588) with a median 6-year follow-up (lines 34–41, 132–140).

The study clearly asks whether higher ERI predicts brain hemorrhage, infarction, and mortality in MHD patients (lines 70–71). The rationale is grounded in prior conflicting evidence but would benefit from citing recent studies.

Ethical approval was obtained (line 254). Inclusion/exclusion criteria are well defined (lines 76–84), and cerebrovascular outcomes were confirmed using imaging, pathology, or other objective clinical evidence criteria (line 91). Multivariate Cox models are appropriately adjusted (lines 122–129; Tables 2–4), and results are supported by Kaplan–Meier analyses (Figures 1–3).

Methods are described clearly, including ERI calculation (line 106), dialysis parameters (lines 85–87), lab and treatment data collection (lines 97–105), and statistical analysis (lines 110–131). The study uses standard statistical tools (SPSS v25.0, line 130). However, the handling of missing data is not discussed, and the Kaplan Meier figures would benefit from number-at-risk tables (Figures 1–3, lines 376–382).

Validity of the findings

The study addresses a well-defined gap in the literature on the relationship between ERI and cerebrovascular events in MHD patients (lines 54–71). The relevance to existing studies is well articulated in the Discussion.

The study uses a large, well-defined cohort (n = 588; line 36) with long-term follow-up (median 6 years; line 42). Analyses are appropriately adjusted via multivariate Cox models (lines 122–129; Tables 2–4), and ERI quartiles are clearly defined (Table 1, line 140). The multivariate models (Tables 2–4) do not adjust for key stroke risk factors such as atrial fibrillation or prior stroke, which may confound results. ERI is assessed only at baseline (line 106).

The Conclusions (lines 234–239) are aligned with the original research question (line 70). Although the study shows that high ERI is associated with increased stroke and mortality risk, the authors do not discuss how ERI could be used in clinical practice.

---

## Round 0.2 · accepted · Accept

· Academic Editor

Accept

Thank you for revising your manuscript to address the concerns of the reviewers. Reviewer 1 now recommends acceptance and I am satisfied that the comments of Reviewer 2 have been addressed. Our production department will reinstate the funding source as discussed, following which, the manuscript will be readied for publication.

Reviewer 1 ·

Basic reporting

The revised version has been further improved.

Experimental design

In the revised version, the method section described in greater detail.

Validity of the findings

No additional comments.

Additional comments

The revised manuscript has been substantially improved and I believe it is worthy of acceptance in this journal.